# High-quality argumentative information in low resources approaches improves counter-narrative generation

**Damián Furman**[1,2]**, Pablo Torres**[3]**, José A. Rodríguez**[3]**,**
**Diego Letzen**[3]**, Vanina Martínez**[4]**, Laura Alonso Alemany**[5]

[1] Departamento de Computación, Universidad de Buenos Aires, Argentina
[2] Consejo Nacional de Investigaciones Científicas y Técnicas, Argentina
[3] Facultad de Filosofía, Universidad Nacional de Córdoba, Argentina
[4] Artificial Intelligence Research Institute (IIIA-CSIC), Barcelona, Spain
[5] Facultad de Matemática, Astronomía y Física, Universidad Nacional de Córdoba, Argentina

## Abstract

It has been shown that high quality fine-tuning boosts the performance of language models, even if the size of the fine-tuning is small. In this work we show how highly targeted fine-tuning improves the task of hate speech counter-narrative generation in user-generated text, even for very small sizes of training (1722 counter-narratives for English and 355 for Spanish). Providing a small subset of examples focusing on single argumentative strategies, together with the argumentative analysis relevant to that strategy, yields counter-narratives that are as satisfactory as providing the whole set of counter-narratives. We also show that a good base model is required for the fine-tuning to have a positive impact. Indeed, for Spanish, the counter-narratives obtained without fine-tuning are mostly unacceptable, and, while fine-tuning improves their overall quality, the performance still remains quite unsatisfactory.

## 1 Introduction

Large Language Models (LLMs) have shown impressive capabilities to generate acceptable texts in a number of scenarios. This has allowed to tackle tasks that were considered too difficult until very recently. Counter-narrative generation in particular is a very challenging task because it requires reasoning, argumentation and world knowledge, in addition to the ability to produce text that is grammatically correct and pragmatically felicitous. Often, counter-narratives resort to pieces of information that are only indirectly implied in the text.

Counter-narrative generation is arising as a valuable application to mitigate some forms of violence in social media. Indeed, automatically generated counter-narratives have been proposed as a primary input to facilitate the task of NGO specialists to counter hate speech (Tekiroğlu et al., 2020).

The predominant strategy adopted so far to counter hate speech in social media is to recognize, block and delete these messages and/or the users that generated it. This strategy has two main disadvantages. The first one is that blocking and deleting may prevent a hate message from spreading, but does not counter its consequences on those who were already reached by it. The second one is that it can generate overblocking or censorship.

An alternative to blocking that has been gaining attention in the last years, is to "*oppose hate content with counter-narratives (i.e. informed textual responses)*" (Benesch, 2014; Chung et al., 2019)[1]. In this way, the consequences of errors in the hate classification are minimized, overblocking is avoided, and it helps to spread a message against hate that can reach people that are not necessarily convinced, or even not involved in the conversation.

However, the huge volume of online hate messages makes the manual generation of counter-narratives an impossible task. In this scenario, automating the generation of counter-narratives is an appealing avenue, but the task poses a great challenge due to the complex linguistic and communicative patterns involved in argumentation.

Traditional machine learning approaches have typically fallen short to address argumentation mining and generation with satisfactory results. However, Large Language Models (LLMs) seem capable of generating satisfactory text for many tasks. Even so, for counter-narrative generation, specific training seems to be required to obtain nuanced, effective counter-narratives. But it is unclear what specific training for this task is, and which information would have a positive effect on this task.

Our hypothesis is that additional information about some argumentative aspects of text will have a positive impact in the quality of the generated counter-narratives.

In this paper, we show that some kinds of argumentative information of hate speech do enhance the proportion of satisfactory counter-narratives

---

[1]No Hate Speech Movement Campaign: http://www. nohatespeechmovement.org/

that a LLM generates. We have focused on the domain of hate speech in social media, and we show that a small dataset of counter-narratives boosts performance, but even smaller subsets have visible impacts in improving performance, if they are enriched with adequate argumentative analysis.

In experiments for English and Spanish, we show that even with very few examples, as is the case for Spanish, highly focused examples and relevant argumentative information do have a positive impact.

The rest of the paper is organized as follows. In the next section, we review relevant work related to automated counter-narrative generation and argumentative analysis of hate speech. Then in Section 3 we describe our dataset of counter-narratives, with which we carry out the comparison of scenarios described in Section 4, where we also describe extensively our approach to the evaluation of generated counter-narratives, based on human judgements, and the prompts used to obtain the counter-narratives. Results analyzed in Section 6 show how fine-tuned LLMs and argumentative information provide better results, which we illustrate with some examples.

## 2 Related work

Automated counter-narrative generation has been recently tackled by leveraging the rapid advances in neural natural language generation. As with most natural language generation tasks in recent years, the basic machine learning approach has been to train or fine-tune a generative neural network with examples specific to the target task.

The CONAN dataset (Chung et al., 2019) is, to our knowledge, the first dataset of hate speech counter-narratives. It has 4078 Hate Speech – Counter Narrative original pairs manually written by NGO operators, translated to three languages: English, French and Italian. Data was augmented using automatic paraphrasing and translations between languages to obtain 15024 final pairs of hate speech – counter-narrative. Unfortunately, this dataset is not representative of social media.

Similar approaches were carried out by Qian et al. (2019) and Ziems et al. (2020). Qian et al. (2019)'s dataset consists of reddit and Gab conversations where Mechanical Turkers identified hate speech and wrote responses. Ziems et al. (2020) did not produce new text, but labeled covid-19 related tweets as hate, counter-speech or neutral based on their hatefulness towards Asians.

In follow-up work to the seminal CONAN work, Tekiroğlu et al. (2020) applied LLMs to assist experts in creating the corpus, with GPT-2 generating a set of counter-narratives for a given hate speech and experts editing and filtering them. Fanton et al. (2021) iteratively refined a LLM where the automatically generated counter-narratives were filtered and post-edited by experts and then fed them to the LLM as further training examples to fine-tune it, in a number of iterations. Bonaldi et al. (2022) apply this same approach to obtain a machine-generated dataset of dialogues between haters and experts in hate countering. As a further enhancement in the LLM-based methodology, Chung et al. (2021b) enhanced the LLM assistance with a knowledge-based retrieval architecture to enrich counter-narrative generation.

Ashida and Komachi (2022) use LLMs for generation with a *prompting* approach, instead of fine-tuning them with manually created or curated examples. They also propose a methodology to evaluate the generated output, based on human evaluation of some samples. This same approach is applied by Vallecillo-Rodríguez et al. (2023) to create a dataset of counter-narratives for Spanish. Both these approaches are targeted to user-generated text, closely related to social media.

However, none of the aforementioned datasets or approaches to counter-narrative generation includes or integrates any additional annotated information apart from the hate message, possibly its context, and its response. That is why we consider an alternative approach that aims to reach generalization not by the sheer number of examples, but by providing a richer analysis of such examples that guide the model in finding adequate generalizations. We believe that information about the argumentative structure of hate speech may be used as constraints to improve automatic counter-narrative generation.

Chung et al. (2021a) address an argumentative aspect of hate speech countering. They classify counter-narratives by type, using a LLM, and showing that knowledge about the type of counter-narratives can be successfully transferred across languages, but they do not use this information to generate counter-narratives.

To our knowledge, ASOHMO (Furman et al., 2023) is the only corpus where tweets of hate speech have been annotated with argumentative information, and CONEAS is the only dataset

of counter-narratives based on the argumentative structure of hate speech. We describe both in the following Section.

## 3 Creating counter-narratives associated to argumentative aspects of hate speech

### 3.1 Argumentative annotation of hate speech

In ASOHMO we enriched the argumentative tweets of Hateval (Basile et al., 2019) with a manual analysis of their argumentative aspects, following an adaptation of the proposal of Wagemans (2016), an analytic approach to represent the semantics of the core schemes proposed by Walton et al. (2008), with fewer categories based on a limited set of general argument features. The following argumentative aspects are manually identified in tweets:

- **Justifications** and **Conclusions**.
- **Type** of Justification and Conclusion: Fact, Policy or Value.
- A **Pivot** signalling the argumentative relation between Justification and Premise.
- Two domain-specific components: the **Collective** which is the target of hate, and the **Property** that is assigned to such Collective.

### 3.2 Counter-narratives based on argument components

In CONEAS (Counter-Narratives Exploiting Argumentative Structure)t[2], each argumentative tweet in ASOHMO is paired with counter-narratives of three different types, defined by applying systematic transformations over argumentative components of the tweet, and a fourth type consisting of any counter-narrative that does not fall under any of the other three.

All counter-narratives, regardless of their type, also follow the guidelines of the Get The Trolls Out project[3]: *don't be aggressive or abusive, don't spread hate yourself, try to de-escalate the conversation, respond thinking on a wider audience than the person posting the original tweet and try to build a narrative.* Annotators were suggested to try to write at least one counter-narrative of each type but only if they came naturally, otherwise they could leave it blank.

The instructions to generate each type of counter-narrative are as follows:

- **Negate Relation Between Justification And Conclusion (Type A)** Negate the implied relation between the justification and the conclusion.
- **Negate association between Collective and Property (type B)** Attack the relation between the property, action or consequence that is being assigned to the targeted group and the targeted group itself.
- **Attack Justification based on it is type (Type C)** If the justification is a fact, then the fact must be put into question or sources must be asked to prove that fact. If it is of type "value", it must be highlighted that the premise is actually an opinion, possibly relativizing it as a xenophobous opinion. If it is a "policy", a counter policy must be provided.
- **Free Counter-Narrative (type D)** All counter-narratives that the annotator comes up with and do not fall within any of the other three types.

An example of each type of counter-narrative can be seen in Figure 1. The dataset consists of a total of 1722 counter-narratives for 725 argumentative tweets in English and 355 counter-narratives for 144 tweets in Spanish (an average of 2.38 and 2.47 per tweet respectively). The annotation process and work conditions are detailed in Appendix A, and the complete guidelines to write counter-narratives can be seen in Appendix C.

## 4 Experimental Settings

We designed a series of experiments to assess the impact of high-quality examples and argumentative information in the automatic generation of counter-narratives via prompting LLMs. We wanted to compare the following configurations:

**Base or XL version of LLM** To assess the impact of the size of the base LLM, we compared the Base (250M parameters) and XL (3B parameters) versions of Flan-T5 (Chung et al., 2021b). Given our limited computational resources, only these two LLMs were compared.

**Finetuned vs Fewshot** To assess the adequacy of different training strategies, we compared a fine-tuning approach and a fewshot approach.

- For fine-tuning, different sets of examples were used.

---

[2]https://github.com/ConeasDataset/CONEAS/
[3]https://getthetrollsout.org/stoppinghate

```
HATE TWEET:
user must deport all illegal migrants india already reeling under constant threat
of muslim radicals curb population
```

**Justification**: `india already reeling under constant threat of muslim radicals curb` `population` **(fact)**
**Conclusion**: `must deport all illegal migrants` **(policy)**
**Collective**: `illegal migrants`
**Property**: `muslim radicals`

**COUNTER NARRATIVE A**  (Negate relation between justification and conclusion)
>    *Deporting illegal migrants will not mitigate the problems with muslim radicals.*

**COUNTER NARRATIVE B**  (Negate relation between collective and property)
>    *Illegal migrants are not necessarily muslim radicals.*

**COUNTER NARRATIVE C**  (Negate justification based on type)
>    *It is not true that India is reeling under threat of muslim radicals.*

**FREE COUNTER NARRATIVE**  (Free)
>    *Deporting illegal migrants without consideration to their circumstances is an inhumane move.*

Figure 1: Examples of each type of counter narratives.

- For few-shot approaches, we tried building prompts with two, five and ten examples from the training partition of the dataset, enclosed in special tokens. We found that if the prompt was built with five or ten examples, the model always generated the same response for all inputs, usually a paraphrase of one of the counter-narratives used as example. So we only report experiments using two examples on the prompt (see figure 4).

**With or without argumentative information**
To assess the impact of argumentative information in the quality of the generated counter-narratives, we enriched the finetuning or fewshot examples with different combinations of the argumentative information available in ASOHMO:

- Collective and Property;
- Justification and Conclusion;
- All argumentative information mentioned above plus the Pivot

**With specific kinds of counter-narratives** To assess the impact of high-quality examples, in the fine-tuning approaches we fine-tuned the base model with different subsets of examples, defined by argumentative criteria, namely, the kind of counter-narrative:

- All counter-narratives

- only counter-narratives of type A (496 training examples in English, 101 in Spanish)
- only counter-narratives of type B (238 in English, 59 in Spanish)
- only counter-narratives of type C (467 in English, 97 in Spanish)

For each type-specific fine-tuning, we compared fine-tuning with the specific subset of counter-narratives only, or enriching them with argumentative information. For each type of counter-narrative, we provided the kind of argumentative information that we deemed relevant for the kind of counter-narrative, that is: Justification and Conclusion for type A, Collective and Property for type B and Justification for type C.

Fewshot experiments were conducted for Flan-T5 Base (small) and XL (larger) models. Fine-tuning was only conducted on Flan-T5 Base because our limited computational resources.

We conducted a small prospective evaluation to find optimal parameters for generation, and we found that using Beam Search with 5 beams yielded the best results, so this is the configuration we used throughout the paper. The details can be seen in Appendix G.

## 4.1 Fine-tuning of the LLM with counter-narratives

To finetune FLAN-T5 with our dataset of counter-narratives, we randomly split our dataset in training, development and test partitions, assuring that all counter-narratives for the same hate tweet are contained into the same partition. Details can be seen on Table 1. All models were trained starting from Flan-T5-Base, in a multilingual setting using mixed English and Spanish examples, with a learning rate of 2e-05 for 8 epochs.

## 5 Evaluation method for generated counter-narratives

Evaluation of counter-narratives is not straightforward. So far, no automatic technique has been found satisfactory for this specific purpose. As cogently argued by Reiter and Belz (2009), superficial n-gram-based metrics like BLEU (Papineni et al., 2002) or ROUGE (Lin, 2004) fail to capture important aspects of generated texts, because they rely strongly on words or n-grams overlap with manually generated examples and they are inadequate in scenarios where there can be many possible good outputs of the model, with significant differences between themselves, such as our case.

Specifically for argumentative texts, Hinton and Wagemans (2023) show that LLMs produce low quality arguments that are nevertheless well-formed utterances. A specific, more nuanced evaluation seems to be necessary to assess relevant features of generated counter-narratives.

Despite the limitations of n-gram-overlap metrics, they are still widely used to evaluate counter-speech (Gupta et al., 2023). But, faced with the shortcomings of these metrics, some authors have conducted manual evaluations for automatically generated counter-narratives, in line with the methods of the Natural Language generation community (Shimorina and Belz, 2022). Manual evaluations typically distinguish different aspects of the adequacy of a given text as a counter-narrative for another. Chung et al. (2021b) evaluate three aspects of the adequacy of counter-narratives: *Suitableness* (if the counter-narrative was suited as a response to the original hate message), *Informativeness* (how specific or generic is the response) and *Intra-coherence* (internal coherence of the counter-narrative regardless of the message it is responding to). Ashida and Komachi (2022), on the other hand, assess these three other aspects: *Offensive-*

*ness*, *Stance* (towards the original tweet) and *Informativeness* (same as Chung et al. (2021b)). Based on these previous works, we manually evaluate the adequacy of counter-narratives in four different aspects (further detail is provided in Appendix D):

- **Offensiveness**: if the tweet is offensive.
- **Stance**: if the tweet supports or counters the specific message of the hate tweet.
- **Informativeness**: Evaluates the complexity and specificity of the generated text.
- **Felicity**: Evaluates if the generated text sounds, by itself, fluent and correct.

Each aspect constitutes a Likert-scale with three possible values, ranging from 1 to 3, where lower values represent the least desirable categories and vice versa. Informativeness is only evaluated if Stance was assigned the higher value possible. Otherwise, it is considered as having a 0 value.

## 5.1 Prospective evaluation of generated counter-narratives

To assess the quality of counter-narratives generated with each approach (with different kinds of argumentative information and LLM settings), we manually evaluated 360 pairs of hate tweet/counter-narratives in English and 180 in Spanish, following the guidelines described in Appendix D. Evaluations are available to the scientific community[4].

We evaluated three random subsets of 20 hate tweets in English and 10 in Spanish. One of the subsets contains only tweets associated with counter-narratives of both types A and C on our dataset, and was used to evaluate models finetuned only with these kinds of counter-narratives. Another contains only tweets associated with counter-narratives of type B and was also used to evaluate models finetuned only with this type of counter-narratives. The last subset contains tweets with counter-narrative pairs of all types, and with this we evaluated the rest of experiments.

For each tweet in the corresponding evaluation subset and for each combination of features to be assessed: fewshot, finetuned, with different kinds of argumentative information and with different sizes of LLM, we generated one counter-narrative. Then, three annotators labeled each pair tweet – counter-narrative according to the four categories described above. Details of the manual evaluation process can be found in Appendix B.

---

[4] https://github.com/ConeasDataset/

| | English | | | | | | Spanish | | | | | |
|---|---|---|---|---|---|---|---|---|---|---|---|---|
| | Tweets | CNs | % corpus | A | B | C | Tweets | CNs | % corpus | A | B | C |
| Train | 509 | 1201 | 69.8% | 496 | 238 | 467 | 105 | 257 | 72.4% | 101 | 59 | 97 |
| Dev | 71 | 173 | 10.0% | 67 | 38 | 68 | 12 | 27 | 7.6% | 12 | 8 | 7 |
| Test | 145 | 348 | 20.2% | 138 | 74 | 136 | 27 | 71 | 20% | 27 | 21 | 23 |
| Proportion of tweets with counter-narrative | | | | | | | | | | | | |
| | | | | 96% | 47% | 90% | | | | 97% | 61% | 89% |

Table 1: Size of dataset partitions of English and Spanish datasets. Columns A, B and C show the amount of counter-narratives used for each partition when training only with counter-narratives of a given type.

### 5.2 A note on the reproducibility of our manual evaluation

As can be expected for such an interpretative task, agreement between judges is far from robust. Thus, the results of this prospective evaluation can only be considered indicative, a first, coarse assessment of the quality of generated counter-narratives. But even with these limitations, the lack of better alternative renders this initial assessment very valuable to determine the impact of different factors, specially argumentative information, in the quality of generated counter-arguments.

Table 2 breaks down the agreement scores between the three annotators, calculated using Cohen's Kappa (Cohen, 1960).

| | 1 vs 2 | 2 vs 3 | 1 vs 3 |
|---|---|---|---|
| Offensiveness | 0.47 | 0.40 | 0.41 |
| Stance | 0.63 | 0.58 | 0.63 |
| Informativeness | 0.49 | 0.42 | 0.54 |
| Felicity | 0.67 | 0.37 | 0.36 |

Table 2: Agreement scores between annotators 1, 2 and 3 using Cohen's Kappa.

In most cases, agreement ranges from Moderate to Substantial (Landis and Koch, 1977). Overall, the category with best agreement is Stance, which seems to be recognizable with more reliability across judges. If we factor out the anomalous judgements of Judge 3, who seems to be having a different understanding of the category, Felicity also seems to be reliably recognized. However, Offensiveness and Informativeness reach lower levels of agreement. Judges do not seem to manage a common concept to assess those categories.

The insights obtained from this first approach will be used to improve the evaluation method, as in Teruel et al. (2018), aiming for more reproducible evaluations. In particular, we will break down the Informativeness category into more fine-grained categories that can capture distinct, relevant aspects of argumentation. We believe that this will provide a more adequate method of evaluation that will enhance the reproduciblity of the evaluation.

### 5.3 Aggregated quality score of generated counter-narratives

Given that there were three judgements for each pair hate tweet – counter-narrative, the final value for each category was determined by majority vote, that is, the value assigned by at least two annotators. When there was no majority vote for a given category, which happened only in 10 cases, we adopted a conservative criterion and assigned the worst of the three possible values.

The final score $S$ for each generated counter-narrative is calculated by summing the values of the four categories of manual evaluation. Since the smallest possible value is 3 and the highest value is 12, we subtracted 2, so the final score $S$ ranges between 1 and 10, as follows:

$$S = \frac{1}{n} \sum_i^n Off_i + Stan_i + Felic_i + Inf_i - 2 \quad (1)$$

where $n$ is the amount of evaluated counter-narratives.

Though we didn't apply an explicit bias when calculating the score, Stance is implicitly more determinant than other categories, since any value lower than 3 for Stance will set the value of Informativeness to 0 automatically.

To facilitate the interpretation of the quality of each approach, we provide another aggregate metric, with discrete values **Excellent**, **Good**, and **None**. Good counter-narratives are those with optimal values for Offensiveness, Stance and Felicity. Excellent counter-narratives are those that also have the optimal value for Informativeness. If values do not meet the criteria for Good, then the counter-narrative is considered None.

| | English | | | Spanish | | |
|---|---|---|---|---|---|---|
| | Score | % Good | % Excellent | Score | % Good | % Excellent |
| Fewshot Approaches | | | | | | |
| Base | 5.70 | 40% | 0% | 2.10 | 0% | 0% |
| Base All | 4.55 | 25% | 5% | 2.20 | 0% | 0% |
| Base Collective | 5.65 | 20% | 5% | 2.20 | 0% | 0% |
| Base Premises | 5.30 | 35% | 5% | 1.90 | 0% | 0% |
| XL | 4.05 | 25% | 0% | 2.80 | 10% | 0% |
| XL All | 2.85 | 10% | 5% | 2.60 | 0% | 0% |
| XL Collective | 3.70 | 15% | 5% | 2.30 | 0% | 0% |
| XL Premises | 2.90 | 0% | 0% | 2.40 | 0% | 0% |
| Finetuned Approaches | | | | | | |
| Base | **7.15** | **60%** | **30%** | 6.60 | 10% | 0% |
| Base All | 5.70 | 25% | 10% | **7.40** | **30%** | 0% |
| Base Collective | **7.40** | 50% | 25% | 6.80 | **30%** | 0% |
| Base Premises | 6.55 | 40% | **30%** | 6.40 | 20% | 0% |
| Base CNs A | 5.75 | 30% | 25% | 4.80 | 0% | 0% |
| Base CNs A Premises | 6.55 | 40% | **40%** | 3.50 | 0% | 0% |
| Base CNs B | 3.05 | 0% | 0% | 2.50 | 0% | 0% |
| Base CNs B Collective | 3.95 | 10% | 10% | 3.20 | 0% | 0% |
| Base CNs C | 4.65 | 10% | 0% | 5.00 | 20% | 0% |
| Base CNs C Premises | 5.60 | 30% | 20% | 3.90 | 0% | 0% |

Table 3: Manual evaluation of automatically generated counter-narratives for English and Spanish hate tweets, using different sizes of the model (Base and XL), two learning techniques (fewshot and finetuning), two different training settings (all counter-narratives or only one kind: A, B or C) and different combinations of argumentative information (no information, Collective and Property, Premises and pivot and all the information available). The score and the proportion of Good and Excellent counter-narratives were calculated by summing the Offensiveness, Stance, Informativeness and Felicity scores assigned by human judges for each manually evaluated counter-narrative, as developed in Section D. The detailed score for each category can be seen in Table 4.

## 6 Analysis of results

Results of the manual evaluation of different strategies for counter-narrative generation for English and Spanish can be seen in Table 3. A summary of this table can be seen in Figure 2, which displays the aggregated proportion of Good and Excellent counter-narratives for each strategy.

In the first place, we can see overall that the results for Spanish are much worse than for English, specially when it comes to using the LLM without fine-tuning, only with a few-shot approach. Using a larger model (XL) has a slightly beneficial effect, but in any case the proportions of Good and Excellent counternarratives obtained by a fewshot approach for Spanish are almost always null.

For English, we can see that the reverse holds: the smaller model (Base) obtains better results for the fewshot approach than the bigger model. Providing argumentative information in this setting seems to negatively affect performance.

Given the poor performance of the XL model in general, and given our limited computational resources, fine-tuning was carried out only for the Base model.

As can be seen in the lower half of Table 3, fine-tuning has a very positive effect in performance. A very valuable conclusion that can be obtained from these results is that, for models that perform well enough, **a small number of high quality examples produce a much bigger improvement in performance than using larger models**, which are also more taxing and even inaccessible for some. The results obtained for Spanish seem to indicate that models require a minimal performance to be improved by additional examples. The question remains, how big the model and the fine-tuning need to be to be able to benefit from the information in examples.

For English, in the left part of Table 3, we can see that the model with the biggest proportion of well-valued counter-narratives (60%), Base, does not

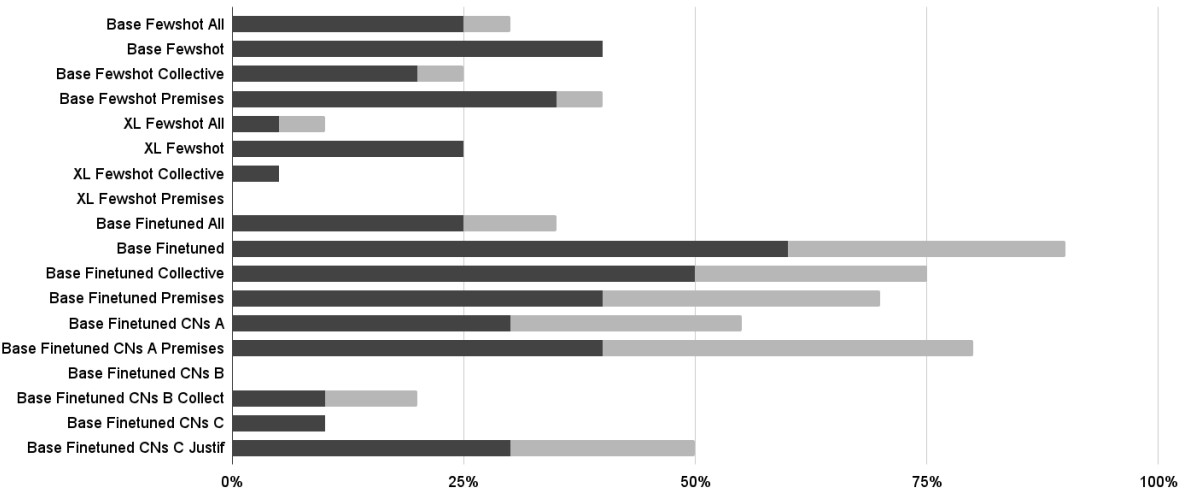

Figure 2: Proportion of Excellent (light gray) and Good (light + dark gray) counter-narratives produced by each approach.

rely on any kind of argumentative information, but some kinds of argumentative information produce comparable results. For example, including information about the Collective targeted by the hate speech yields 50% well-valued counter-narratives, and a higher overall score.

Moreover, in qualitative inspection we have observed that counter-narratives produced by providing argumentative information tend to be more informative, as can be seen in Figure 3. However, our current evaluation method is not fine-grained enough to reflect these subtle differences in quality. Future developments of the evaluation method will aim to capture such differences.

When we fine-tune the model only with counter-narratives of a given type (lower quarter of Table 3), the number of examples for fine-tuning is much smaller, but performance is not proportionally affected. Although the proportion of counter-narratives that are valued as Good is lower, the proportion of Excellent increases, especially for counter-narratives of type A (negation of the relation between Justification and Conclusion) for which information about the Justification and Conclusion is provided, even though the training examples are less than half (1201 training examples for the whole corpus vs. 496 examples of counter-narrative A).

For counter-narratives of type B and C, however, improvements are not as good. For counter-narratives of type B, the small number of examples may be the reason for such a small improvement in performance. For counter-narratives of type C

(negation of Justification), with almost as many examples as in type A narratives, the smaller ratio of improvement may be attributed to more complex argumentative structures.

In any case, providing the relevant argumentative information, Collective for type B and Premises for type C, always produces an improvement. In fact, even when finetuning with all types of counter-narratives, providing information about the Collective and Premises yields a improvement higher than providing all available argumentative information, but not as good as not providing any argumentative information at all.

**For Spanish**, the performance of the model without specific training is much worse than for English, even with fine-tuning, producing almost no Good and Excellent counter-narratives. However, when the model is fine-tuned with counter-narratives, even if the number of training examples is very small (257 in total), the improvement in the performance of the model is notable, even much higher than for the case of English. Indeed, while English goes from a score of 5.70 for the base model with few shot to a score of 7.15 for fine-tuned, Spanish goes from 2.1 to 6.6, obtaining a much bigger improvement and coming much closer to the performance of the model for English. The improvement is much more notable when argumentative information is provided, even reaching the score for the best approaches for English (7.4) when all argumentative information is provided. However, the proportion of Good narratives remains low, and Excellent is nonexistent.

| Tweet with argumentative information: | Tweet without argumentative information: |
|---|---|
| street interview whit italians "send all migrants back to where they came from they block streets to pray " - free speech time - :link: @user | **Justification**: street interview whit italians "send all migrants back to where they came from they block streets to pray " (fact) | **Conclusion**: "send all migrants back to where they came from they block streets to pray " (policy) | **Pivot**: migrants - they - they | street interview whit italians "send all migrants back to where they came from they block streets to pray " - free speech time - :link: @user |
| Counter-narrative: | Counter-narrative: |
| I don't think it's a good idea to send all migrants back to where they came from. | I don't think it's the right thing to do. |

Figure 3: Counter-narratives obtained for the same tweet with different strategies: including argumentative information (left) and without argumentative information (right).

When fine-tuned with types of counter-narratives separately, improvements are smaller, probably due to the very small number of training examples.

In any case, the evaluation we carried out is small, mostly indicative, and these tendencies will be further explored in future work.

## 7 Conclusions and future work

We have presented an approach to generate counter-narratives against hate speech in social media by prompting large language models with information about some argumentative aspects of the original hate speech.

We have shown that finetuning a smaller model with a small corpus of high-quality examples of pairs hate speech – counter-narrative yields a bigger improvement in performance than using larger language models.

We have also shown that some kinds of argumentative information do have a positive impact in generating more specific, more informative counter-narratives. In particular, we have found that the types of counter-narrative that negate the relation between the Justification and the Conclusion and that negate the Justification have an improvement in performance if argumentative information about the Justification and the Conclusion is provided.

Moreover, we have also found that argumentative information makes a positive impact in scenarios with very few tweets, as shown by our experiments for Spanish. Although the quality of the counter-narratives generated for Spanish is lower than for English, it is encouraging that argumentative information has a much higher positive impact than for a scenario with bigger models and more training examples, and we will continue to annotate examples for Spanish to improve the generation of counter-narratives.

We will also explore other aspects of the quality of counter-narratives, with a more insightful, more extensive human evaluation. We will also explore the interaction between argumentative information and other aspects, like lexic, level of formality, and culture.

## Acknowledgments

This work was supported by funds provided by Agencia Nacional de Promoción Científica y Tecnológica under grants PICT-2018-0475 (PRH-2014-0007) and PICT-2020-SERIEA-01481, and CONICET under grant PIP 11220200101408CO. The authors also acknowledge support by the Spanish project PID2022-139835NB-C21 funded by MCIN/AEI/10.13039/501100011033, MCIN/AEI/10.13039/501100011033 and by "European Union NextGenerationEU/PRTR". The development of the CONEAS corpus of counternarratives was supported by NAACL as part of the ContraHate challenge[5]. This work used computational resources from CCAD – UNC (https://ccad.unc.edu.ar/), which are part of SNCAD – MinCyT, Argentina.

---

[5]https://sites.google.com/unc.edu.ar/contrahate

## Limitations and Ethical Considerations

In the first place, we would like to make it clear that the counter-narratives created for this study are the result of the subjectivity of the annotators. Although they have been instructed through a manual and training sessions, counter-narrative generation is an open-ended task, subject to great variability between persons. Even if we have taken care to avoid it, it can still be the case that the cultural and personal biases of the people generating the counter-narratives may produce some texts that result offensive for some of the collectives involved, or that do not share values with some parts of society.

Also, it is important to note that the automatic procedures obtained are prone to error, and should not be used blindly, but critically, with attention to possible mistakes and how they may affect users, groups and society.

Then, it is also important to note that the corpus used for this research is very small, specially in the Spanish part, so the results presented in this paper need to be considered indicative. A bigger sample should be obtained and annotated to obtain more statistically significant results.

The findings of this research can potentially inform the development and improvement of language models and chatbot systems. However, we emphasize the importance of responsible use and application of our findings. It is essential to ensure that the identified argumentative components are utilized in a manner that promotes reasoned usage and does not contribute to the spread of hate speech or harmful rhetoric. We encourage researchers and developers to consider the ethical implications and societal impact of incorporating argumentative analysis into their systems.

The data have been adequately anonymized by the original creators of the Hateval corpus.

Studying hate speech involves analyzing and processing content that may be offensive, harmful, or otherwise objectionable. We acknowledge the potential impact of working with such content and have taken steps to ensure the well-being of the research team involved. We have provided comprehensive guidelines and training to our annotators to mitigate any potential emotional distress or harm that may arise from exposure to hate speech. Additionally, we have implemented strict measures to prevent the dissemination or further propagation of hate speech during the research process.

Finally, we have not specifically conducted a study on biases within the corpus, the annotation or the automatic procedures inferred from it, nor on the LLMs that have been applied. We warn researchers using these tools and resources that they may find unchecked biases, and encourage further research in characterizing them.

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

## A Details of the annotation process for the CONEAS corpus

Manual generation of CONEAS counter-narratives was conducted by two annotators, a Computer Science PhD student and an undergraduate Philosophy student. To begin with, annotation guidelines were created by the research team, including annotators, describing how to use the argumentative analysis of hate tweets to write counter-narratives using different strategies (A, B and C), as seen in Appendix C. Then each annotator wrote one counter-narrative of each type for the same subsample of 20 tweets in English, then compared their annotations and discussed how to consolidate the counter-narrative writing process to improve:

- Adequacy of the counter-narrative as an acceptable, tailored reply for a given hate tweet;
- Adequacy to the specific type of counter-narrative;
- Robustness of the written narrative.

This process is challenging since the task is inherently subjective and variability between annotators does not necessarily imply that one of them is wrong, but that guidelines need to cover such variability, without being too lax or too vague. Then, 50 new tweets were annotated, and difficult issues were discussed to achieve a better alignment between annotators, but without a need to update the guidelines. After that, they wrote counter-narratives for all the rest of the dataset. After the whole annotation, a third annotator reviewed 50 random pairs of hate tweet/counter-narrative using the guidelines as reference, without finding major inadequacies.

The first annotator labeled 20% of the total dataset and was not paid specifically for this task, as it is part of his own reserach project. The second annotator labeled the other 80% and was paid 25USD/hour for 80 hours of work, partially funded by NAACL as part of the organization of the ContraHate challenge[6].

## B Details of the manual evaluation process

For the manual evaluation of the hate tweet – counter-narratives pairs (360 pairs in English and 180 in Spanish), there were three annotators: two

[6]https://sites.google.com/unc.edu.ar/contrahate

of the authors of the paper and a PhD in an unrelated field without previous acquaintance with the task. The two authors were unpaid, the unrelated judge was paid 25U$D/hour. In total, each annotator labeled 540 hate tweet/counter-narrative pairs. A total of 5 hours was devoted to this evaluation, summing up all 3 evaluators.

## C Guidelines to create counter-narratives

The following are the instructions for annotators to create counter-narratives for hate tweets against immigrants within the HatEval corpus (Basile et al., 2019), enriched with argumentative annotations in ASOHMO (Furman et al., 2023):

Three types of counter narratives must be written, based on the argumentative annotation of ASOHMO. Each type of counter narrative will propose a specific hint for the annotator to follow but they must be written to sound like a natural spontaneous response on Twitter. All three types of counter-narratives must be written unless it is not possible to write something coherent using the given hint.

It might be possible that two counter-narratives written following different hints end up being very similar. This is not a concern and in any case, the annotator will try whenever it is possible to emphasize different aspects of the counter narrative to make them be the least similar possible.

Counter narratives should be written following the guidelines of the Get The Trolls Out project: *don't be aggressive or abusive, don't spread hate yourself, try to de-escalate the conversation, respond thinking on a wider audience than the person posting the original tweet and try to build a narrative* (see your answers as a long term, repetitive process).

If you think that there is a strong counter-narrative that doesn't fit any of the three proposed types, add it as a fourth type (CounterNarrativeD).

### C.1 Negate relation between justification and conclusion (CounterNarrativeA)

The basic idea for this type of counter narrative is to negate the relation between the conclusion and the justification, therefore negating the core of the user's reasoning. In most cases the reason given by the user is forced or fallacious and there is a direct counter-narrative. However, in some cases it's necessary to give an indirect response or to concentrate on a particular aspect of the reasoning

that is more vulnerable to an attack.

> Example:
> *We should deport all illegal immigrants because they are breaking the law and the law says they get deported.*

In this case, directly negating the relation between conclusion and justification would result in something like "just because the law says something doesn't mean you have to do it" but it's not a very convincing narrative. If, on the other hand, the perspective is changed, the counter-narrative could state that "Just because the law says something, it doesn't mean it's right" or "Laws can change if they don't adjust to the reality of innocent people suffering. What should prevail is our humanity and our will to help the needed and not our will to get rid of them".

### C.2 Negate relation between collective and properties associated to them (CounterNarrativeB)

The core of this type of counter-narrative is to negate the relationship between the collective being attacked and the properties, actions or consequences related to them. If no collective or property were annotated, you can skip this counter-narrative.

### C.3 Attack the justification of the claim (CounterNarrativeC)

The third type of counter-narrative consists of refuting the justification of the argument. This refuting must be done considering the type of the justification premise. If it is of type "fact", then the fact must be put into question or sources must be asked to prove that fact. If it is of type "value", the characteristic of the premise of being an opinion must be highlighted and the speaker's opinion itself must be relativized as a xenophobous opinion. If it is a "policy" a counter or opposite policy must be answered.

### C.4 Free Counter-narrative (CounterNarrativeD)

If and only if there is a strong counter-narrative that doesn't fit on any of the types proposed, the annotator can add it as a free counter-narrative type.

## D Guidelines for manual evaluation of counter-narratives

We manually evaluate[7] the adequacy of counter-narratives in four different aspects:

- **Offensiveness**: if the tweet is offensive to either the target group, the author of the tweet or any other group or person. Possible values are: Offensive; Possibly Offensive/Not clear; Not offensive.
- **Stance**: if the tweet supports or counters the specific message of the hate tweet. Possible values are: Supports the original message; Not clear/Changes subject wrt original tweet; Counters the original message. Stance incorporates a certain notion of suitableness, since it assigns value "Changes the subject" if the counter-narrative is not responding specifically to the standpoint of the original tweet.
- **Informativeness**: Evaluates the complexity and specificity of the generated text. Only counter-narratives with a "Counters" Stance are evaluated. Possible values are:

  1. **Generic statement**: replies that don't incorporate any information mentioned on the tweet and could counter many different hate messages (e.g "I don't think so" or "That is not true").
  2. **Specific but not argumentative**: the reply is a simple statement, possibly composed of a single sentence without providing justification for the stance but referring to some specific aspect of the original tweet. Usually they comply with a formula composed of a prefix (like "I don't think that" or "Do you have proof that") and a verbatim copy of some part of the hate tweet.
  3. **Specific and Argumentative**: counter-narratives with some degree of elaboration of the information contained on the hate message. We identified three common patterns that we associate with this value:

     A - replies that take more than one element from the original message and establish some relation between them (e.g. "I don't see the relation between

---

[7]Results of the evaluation can be found on https://shorturl.at/aetFZ

{*element from the original message*} and {*other element from the original message*}").

B - A simple statement declaring stance over a single element from the original tweet but adding a second coordinated statement with personal appreciations about it (e.g. "I don't think we should {*some policy mentioned on the tweet*}. It is a bad idea").

C - An argumentative reply that necessarily inferred some information not mentioned explicitly on the original tweet showing a comprehensive understanding of the meaning of the hate message (e.g. a reply to a tweet concluding with #BuildTheWall saying "Building a wall would cost more to the taxpayers" or "Building a wall won't give you more control over illegal trafficking").

- **Felicity**: This category is related to Chung et al. (2021b)'s Intra-Coherence, but also considering additional dimensions like syntactical and semantic correctness. It evaluates independently of the original tweet, if the generated text sound, by itself, fluent and correct. There are three possible values: The text is incoherent or semantic or syntactically incorrect; The text is coherent with small errors like in coordination of genre/tense/etc. or repeating parts of the original text without adapting them to the text being generated; The text is fluent and sounds correct.

## E   Breaking down the score calculation

Table 4 shows how the score assigned to each batch of HS-CN pairs corresponding to each experimental settings is broken down into the four categories used for the manual evaluation: Offensiveness, Stance, Informativeness and Felicity.

## F   Prompt for fewshot approaches

In Figure 4 an example prompt for the fewshot approach can be seen.

## G   Generation settings

We conducted an independent experiment to find the best performant decoding method for the generation of counter-narratives. We finetuned a FLAN-T5 Base model with ASOHMO without using the

```
"Generate a non-hateful response for
the following hate messages:

<SHS>unbelievable census bureau:  44
6% in ca dont hashtag speakenglish at
home 35 6% 35 6% in tx 34 5% in nm
31 7% in nj 31 0% in ny our schools r
getting 150% funding for every student
who doesnt speak hashtag english well
parents job hashtag stop immigration
https://t co/vi6xxhik1b<EHS> => <SCN>
And would ending immigration improve
English? I don't think so... <ECN>

<SHS>100     bangladeshi    nationals
apprehended   near   texas   border   in
3 weeks via @user hashtag noamnesty
hashtag   deport   illegals   hashtag
buildthatwall  hashtag  trump  hashtag
maga hashtag masa<EHS> => <SCN>Do you
know why where they apprehended? Why do
you assume they were illegals? <ECN>

<SHS>" + Tweet + "<EHS> => <SCN>"
```

Figure 4: Prompt used for English fewshot counter-narratives generation.

argumentative information. We then generated 10 counter-narratives for 10 random tweets in English that were not among the 20 used for the main evaluation using four decoding strategies: Greedy Search without sampling, with sampling using a temperature value of 0.7, with sampling using Top P with a p value of 0.92 and with Beam Search using 5 beams. Results of this evaluation are shown in Table 5. Results show that Beam Search outperforms the rest of the decoding strategies explored, so we used this generation method for all experiments in this work. A manual examination of the examples generated shows that beam search produces less diverse results, repeating some words or phrases between replies to different hate tweets. Sampling techniques produce more diverse responses, but are also more prone to generate incoherent phrases.

| | English | | | | | Spanish | | | | |
|---|---|---|---|---|---|---|---|---|---|---|
| | Off | Stance | Infor | Fel | Score | Off | Stance | Infor | Fel | Score |
| Fewshot Approaches | | | | | | | | | | |
| Base | 2.50 | 2.25 | 0.40 | 2.55 | 5.70 | 1.30 | 1.30 | 0.00 | 1.50 | 2.10 |
| Base All | 1.95 | 1.80 | 0.45 | 2.35 | 4.55 | 1.30 | 1.20 | 0.00 | 1.70 | 2.20 |
| Base Collective | 2.45 | 2.05 | 0.30 | **2.85** | 5.65 | 1.20 | 1.20 | 0.00 | 1.80 | 2.20 |
| Base Premises | 2.05 | 2.05 | 0.55 | 2.65 | 5.30 | 1.00 | 1.00 | 0.00 | 1.90 | 1.90 |
| XL | 1.65 | 1.65 | 0.40 | 2.35 | 4.05 | 1.40 | 1.40 | 0.10 | 1.90 | 2.80 |
| XL All | 1.30 | 1.30 | 0.15 | 2.10 | 2.85 | 1.40 | 1.30 | 0.00 | 1.90 | 2.60 |
| XL Collective | 1.70 | 1.60 | 0.25 | 2.15 | 3.70 | 1.10 | 1.10 | 0.00 | **2.10** | 2.30 |
| XL Premises | 1.55 | 1.40 | 0.00 | 1.95 | 2.90 | 1.30 | 1.20 | 0.00 | 1.90 | 2.40 |
| Finetuned Approaches | | | | | | | | | | |
| Base | **2.55** | 2.55 | 1.40 | 2.65 | 7.15 | 2.60 | 2.60 | **1.70** | 1.70 | 6.60 |
| Base All | 2.30 | 2.15 | 0.60 | 2.65 | 5.70 | 2.90 | **2.80** | 1.60 | **2.10** | **7.40** |
| Base Collective | **2.55** | **2.60** | **1.45** | 2.80 | **7.40** | 2.70 | 2.70 | 1.40 | 2.00 | 6.80 |
| Base Premises | 2.30 | 2.35 | 1.20 | 2.70 | 6.55 | 2.60 | 2.70 | 1.40 | 1.70 | 6.40 |
| Base CNs A | 2.35 | 2.25 | 0.90 | 2.25 | 5.75 | 2.50 | 2.30 | 0.60 | 1.40 | 4.80 |
| Base CNs A Premises | 2.50 | 2.35 | 1.30 | 2.40 | 6.55 | 2.20 | 2.00 | 0.20 | 1.10 | 3.5 |
| Base CNs B | 1.90 | 1.80 | 0.05 | 1.30 | 3.05 | 1.70 | 1.80 | 0.00 | 1.00 | 2.50 |
| Base CNs B Collective | 2.15 | 2.15 | 0.35 | 1.30 | 3.95 | 2.00 | 2.00 | 0.20 | 1.00 | 3.20 |
| Base CNs C | 2.50 | 2.20 | 0.35 | 1.60 | 4.65 | **3.00** | 2.30 | 0.3 | 1.40 | 5.00 |
| Base CNs C Premises | 2.20 | 2.25 | 0.85 | 2.30 | 5.60 | 2.20 | 2.10 | 0.30 | 1.30 | 3.90 |

Table 4: Score assigned to counter-narratives generated by each experimental setting, broken down by each of the four categories used for evaluation: **Off**ensiveness, **Stance**, **Infor**mativeness and **Fel**icity. **Off**ensiveness, **Stance** and **Fel**icity range between 1 and 3, while **Infor**mativeness ranges between 0 and 3. The Score is the sum of the four previous categories but subtracting 2, so it ranges between 1 and 10.

| | Ofensiveness | | | Stance | | | Informativeness | | | Felicity | | | Good | Excellent |
|---|---|---|---|---|---|---|---|---|---|---|---|---|---|---|
| | 1 | 2 | 3 | 1 | 2 | 3 | 1 | 2 | 3 | 1 | 2 | 3 | | |
| Greedy | 10% | 20% | 70% | 0% | 40% | 60% | 10% | 20% | 30% | 10% | 40 % | 50% | 30% | 10% |
| Greedy w/Temp | 20% | 30% | 50% | 20% | 60% | 20% | 0% | 10% | 10% | 40% | 10 % | 50% | 10% | 10% |
| Greedy w/Top P | 80% | 0% | 20% | 40% | 20% | 40% | 10% | 0% | 30% | 40% | 0 % | 60% | 20% | 10% |
| Beam P | 10% | 0% | **90%** | 10% | 0% | **90%** | 0% | 30% | **60%** | 0% | 30 % | **70%** | **70%** | **50%** |

Table 5: Results for manual evaluation of different decoding strategies.

| | Score | % Good | % Excellent |
|---|---|---|---|
| Greedy | 7.00 | 30% | 10% |
| Greedy w/Temp | 4.60 | 20% | 10% |
| Greedy w/Top p | 4.90 | 20% | 10% |
| Beam Search | **8.70** | **70%** | **50%** |

Table 6: Scores for different generation strategies using a finetuned Flan T5-Base model.