# OpenReview forum: "High-quality argumentative information in low resources approaches improve counter-narrative generation"
_EMNLP/2023/Conference — EMNLP 2023 Findings_

### Official Review · Reviewer_GdEP · 2023-08-01

**Soundness:** 3

**Excitement:**

3: Ambivalent: It has merits (e.g., it reports state-of-the-art results, the idea is nice), but there are key weaknesses (e.g., it describes incremental work), and it can significantly benefit from another round of revision. However, I won't object to accepting it if my co-reviewers champion it.

**Paper Topic And Main Contributions:**

The paper describes work on the addition of human-annotated argumentative aspect information to the standard finetuning and few-shot learning to train generative models to combat hate speech by generating counter-narratives. The paper derives its motivation from the fact that current approaches in the field only ever make use of the actual hate speech text itself for finetuning, and no one has yet explored on incorporating context in the form of extracting argumentative information and using this in for training models. The paper then collects four types of argumentative information from texts, including negations between justification and conclusion, property and collective, attack justification, and free counter-narrative. The paper explores training only on a very limited set of models and experimental procedures, citing concerns on carbon footprint and limited computational resources. Results show that the use of additional argumentative information does improve performances even for languages such as Spanish with relatively small datasets compared to English and that smaller models have produced more favorable effectivity than larger models for the task.


**Questions For The Authors:**

Please see questions and concerns from the weaknesses section and address them.


**Reasons To Accept:**

The paper tackles a very important application of NLP in a real-world scenario which is the generation of counter-narratives against hate speech (and fake news probably as well). The motivation is clear, and readers would easily appreciate the efforts of the authors done for the study. The paper is also relatively well written (but requires some level of structuring and reorganization to make it more readable to the wider audience) and provides an acceptable level of analysis of results. More than that, I believe two of the datasets constructed for this study, the CONEAS data plus the additional ground truth dataset used for testing, would be beneficial to the NLP community.


**Reasons To Reject:**

While there are definitely strong points to the paper, I still have some concerns about the technical rigor and how the paper details some of its experimental procedures. Addressing these would help improve the quality of the work.

1. The annotation process and the annotators should be described in more detail, including reliability testing, payment, and background qualifications--especially since subjectivity is one major concern for the annotation of these types of tasks. The annotation process should be clearly described, including the instructions given to the annotators and the quality control measures that were used. This information was provided for the creation of the ground truth dataset, but not for the construction of the CONEAS dataset.

2. The proposed few-shot learning seems to be sensible for the study, but the experiment done for the paper seems to be very limited to me. If only two examples are used at a time, the model will not be able to learn as much as it could if more examples were available. I believe it would be interesting to see if there are positive changes in the model's performance as more or fewer examples are given to it during training. Especially if, in the discussion section, the authors highlighted some improvements in using a smaller T5 model compared to the XL one.

3. The authors seem to be overusing the claim of reducing carbon footprints to not perform further experiments to further ground the findings. While this reason may be noble in some sense, I don’t find it as an acceptable excuse to limit the freedom of the authors to do more training that will support their results. Especially for not finetuning Flan-T5 XL which leaves the whole experiment recipe to be incomplete.

4. There should be justification as to why Flan-T5 was used for the task compared to other existing generative models that are easy to finetune such as GPT2. Is there some special feature with Flan-T5 that makes it better for the task?


**Reproducibility:**

2: Would be hard pressed to reproduce the results. The contribution depends on data that are simply not available outside the author's institution or consortium; not enough details are provided.

**Reviewer Confidence:**

4: Quite sure. I tried to check the important points carefully. It's unlikely, though conceivable, that I missed something that should affect my ratings.

**Typos Grammar Style And Presentation Improvements:**

The title is grammatically incorrect. It sounds more acceptable as “low resource approaches” than “low resources approaches”.

It might be better for the EMNLP audience as well as general readers for the authors to formalize the scoring mechanism developed in a mathematical formula. I suggest this as this can serve as one of the main contributions of the paper. The impression I’m getting from the scoring is that there may be two formulas to be added since it was mentioned in the criteria that for Informativeness, only counter-narratives with Stance are evaluated. All of these should be properly formalized to avoid confusion.

In Section 4.3, I suggest adding one table containing the details of the new ground truth dataset created. Maybe add this in the Appendix just to have another reference for readers who need quick details about your dataset.

I appreciate that the authors provided in-depth discussion from the results they obtained in Table 3. However, condensing all of these in one long paragraph reduces the readability of the discussion. I suggest the authors to add subsections to split the different parts of discussions into something that in targets from the experiment. For examples: Subsection A. Effect of Model Size to Performance, Subsection B. Fewshot vs. Finetuning, etc.

---

> ### Author Rebuttal · Authors · 2023-08-29
>
> [this is a general comment for all three reviewers] We appreciate the effort made by the reviewers to provide insightful feedback on our work. We understand that it does not exactly conform to the standard experimental paper with standard datasets and standard metrics. We made the decision to address questions that are not straightforward incremental on previous work, and we understand this requires an extra effort from reviewers to identify strong and weak points. We are very thankful for this effort and the very valuable suggestions that they provide. We will be sure to incorporate as much as we can of their work in the final version of this paper.
>
> We appreciate the suggestions to improve the presentation, find them very pertinent and will incorporate them in the camera-ready version of the paper if accepted.
>
> With respect to the fact that only results with Flan-T5 were reported, we have to describe our research context. We work in a low-income country, with a very small budget to carry out this and mostly any research project. All the resources used to run our experiments come from an HPC center in a public university, which granted us access to a cluster with one NVIDIA A10 and one NVIDIA A30, each with 24GB RAM each, and we had to go through a selection process that was time consuming to be granted this access. To be honest with the reviewer (to whom we highly appreciate all the work taken to review our work), one of the reasons (besides the carbon footprint) why we didn’t finetune Flan-T5-XL is because it doesn’t fit on the RAM memory we have available if we want to fine-tune it (if we only want to generate text it does fit, and the same thing happened with FLAN-T5-Large). We have to constantly make decisions about what is the best way to spend the funds we’ve got. In this case, we decided that it was better to spend it on developing the most amount of data of the best quality possible and then use the models that would fit in the cluster we have access to instead of buying computing resources or devoting our time to write proposals to obtain access to computing resources provided pro bono by other institutions. Although we agree that it would be interesting to fine-tune Flan-T5-XL, we think that one advantage of this experiment settings is that they allow the experiment to be reproduced without the need of costly hardware. One main standpoint of our work is that quality data is preferred over bigger models, so if future researchers with limited funds like us have to choose between either spending on annotation or spending on hardware we can provide evidence to support the first. To prove that, we believe that comparing a finetuned base model against a non-finetuned large model is sufficient.
>
> However, we did experiment with different architectures of models of smaller sizes. To be able to conduct our experiment according to the four categories of analysis that we proposed we needed models with at least two sizes available: a base size small enough to be finetuned without an excessive computational effort (around or less than 500M params) and a larger version used for comparison. We discarded all models that didn’t have a version lighter than 3b parameters such as Llama and that were not OpenSource, like ChatGPT and GPT4. We decided to only use open-source models available for the research community that allows us to reproduce the experiment afterwards. We chose three models complying with these specifications: Bloom (560M and 3B parameters versions), GPT2 (137M and 1.5B parameters versions) and Flan-T5 (base-248M and XL-3b parameters). We conducted small experiments with each of these models, generating counter-narratives for 20 random English tweets using a few-shot approach. We carried out a manual inspection of the results and found that Flan-T5 produced the most coherent responses by far, at least using this approach and for the base model. Besides that, only Flan-T5 and Bloom are multilingual, and we were interested in the approach working straightforwardly for Spanish. We didn’t include these results because of space constraints but mostly because we didn’t evaluate these results thoroughly using our proposed four categories of analysis. We agree with the reviewer, however, that it is important to add justification about why we chose the model that we used for our experiments and we will add both a description of the comparison we did between those models and the evaluation of the generated results using our four categories of analysis for the camera-ready if our work is accepted.
>
> [This is a shared response for the three reviewers] Thanks to your comments we understand that the values of the score are difficult to interpret. In the final version of the paper we will provide the algorithm showing how the score is calculated, and will illustrate it with a couple of examples in an Appendix. We believe that a formula will not make the meaning of the score more clear, because it is a sum of the score obtained for each of the four dimensions making up the final score, which are each a Likert scale ranging from 1 to 3 in three cases and 0 to 3 in the fourth, with a maximum possible score of 12. Moreover, one of those dimensions, “Informativeness” only scores more than 0 if the dimension of “Stance” scores at the maximum of 3.
>
> Normalizing the final score to a range from 0 to 1 is definitely a suggestion that we will take, as it makes the score more interpretable. However, seeing the difficulties in interpreting the metric, we will also provide the detail of the score obtained by each approach for each of the four dimensions evaluated, as we believe this provides more transparency for the way the final score is built up.
>
> [This is a shared response for two reviewers] Thank you for pointing out the missing information about the annotation process and the annotators. We will include it in the camera-ready version of the paper, in case it is accepted. The details are as follows:
> Manual generation of CONEAS counter-narratives was conducted by two annotators in three phases. The annotators are both from academic backgrounds, one being a Computer Science PhD student and the other, an undergraduate Philosophy student within the area of Theory of Argumentation. Annotation guidelines were created by the research team before the annotation process began, describing how to use the argumentative analysis of hate tweets to write counter-narratives using different strategies (A, B and C). Both annotators participated on a two hour training to get familiar with the guidelines. Then, during the first phase, each annotator wrote one counter-narrative of each type for the same subsample of 20 tweets in English, then compared the results of their annotation and discussed among each other on how to improve the counter-narrative writing process according to three criteria: a - Adequacy of the counter-narrative as an acceptable, tailored reply for a given hate message; b - Adequacy of the counter-narrative to the specific type of counter-narrative; c - Robustness of the written narrative. This process is challenging since the task is inherently subjective and variability between annotators does not necessarily imply that one of them is wrong, but that guidelines need to cover such variability, without being too lax or too vague. After re-writing and consolidating the guidelines, some counter-narratives were re-written according to the updated guidelines. The second phase of the annotation process consisted in annotating 50 new tweets, and discussing issues that may have arisen and some difficult cases, just to achieve a better alignment between annotators, but without a need to update the guidelines. The third phase consisted on writing counter-narratives for all the rest of the dataset. After this third step, a third annotator reviewed 50 random pairs of hate tweet/counter-narrative using the guidelines as reference, without finding major inadequacies. The first annotator labeled 20% of the total dataset and was not paid (he is part of our research team) while the second annotator labeled the other 80% and was paid 25U$D/hour for 80 hours of work. This project was partially funded by NAACL (we plan to add an acknowledgment and sincere thanks for this in the camera ready version of the paper, if accepted). We want to thank the reviewer for pointing this out. We had originally omitted this information because of space constraints in the paper, but will include it even if it is necessary to add an Appendix.
>
> Concerning the manual evaluation of the automatically generated counter-narratives, this is in a first stage of consolidation. We are aware that evaluation of counter-narratives is a highly complex task, and we intend to carry out a more extensive evaluation with the lessons learned from this first experience. That is why this first phase was small. A small set of guidelines was developed. The annotators were the two annotators who created the manual counter-narratives and a third annotator who is also invested in the project, went through the training with the evaluation guidelines and also participated in the discussion of some test examples to build common ground between annotators. For this work, annotators were paid  the same rate per hour as for the creation of the counter-narratives, and a total of 5 hours was devoted to this evaluation, summing up all 3 evaluators.
>
> Despite the NAACL grant, we have a low budget to carry out this project and had to decide what to fund. We decided to spend most of our budget in obtaining a high quality annotated dataset, instead of investing in obtaining more powerful hardware, because we believe that the quality of annotations makes a bigger impact in this kind of problem. Our main research question was to assess whether this kind of high quality argumentative information did have an impact in the performance of models, more than extensive comparison of models, so we invested most of our little budget in obtaining a high quality dataset. We believe that, to obtain a high quality dataset, one has to employ properly educated annotators, provide them with adequate training and revision processes, and provide high quality working conditions, with good pay and as little as possible time pressure.
>
> Moreover, we also believe that, now that the dataset is publicly available, experiments with larger language models, requiring more powerful computing, can also be carried out by other teams with more computing power available.
>
> Concerning how limited the few-shot learning approach was, we actually conducted experiments for the few-shot approach providing more examples in the prompt for the Flan-T5 model. In particular, we tried prompts with 3, 5 and 10 examples. In all cases, both for base and XL models, when 5 or more examples are used, the text generated by the model is always the same no matter the input taken. In these cases, the model reproduces, either verbatim or with small variations, one of the counter-narratives defined on the examples used on the prompt. Although in some cases, the counter-narrative generated by the model could be considered more or less accurate, we consider this to be mere chance, since the model always returns the same, no matter the input. We hypothesized that this behavior might be due to the model falling in a local minimum. To force the model to produce variability in the output, we ran a small test using different generation settings (particularly, using top-sampling and temperature) and checked that in these cases there is higher variation among the responses, but the quality of the generated text was noticeably worse. Since we couldn’t devote the time to properly understand the underlying reason for this anti-intuitive behavior, and considering that manual evaluation of the generated counter-narratives is costly, we decided not to evaluate these cases, and because of space constraints, we didn’t include a description of this phenomenon in the paper.
>
> We chose to use only two examples and low temperature considering that in a preliminary observation, this seemed to be the number that yielded good but also diverse replies. However, we understand now with the help of the comments made by the reviewer, that it is important to report this and we are planning on doing so. We will describe the experiments we carried out with more examples in the prompt, showing the average distance between the sentence embedding of one generated counter-narrative to the sentence embedding of all the rest. Experiments where all the generated responses are the same will have an average distance of zero, indicating that the model always returns the same. This indicator will be useful to motivate the choice of n=2 for the number of examples provided in the prompt.
>
> [This is a shared response for two reviewers] To our understanding, the paper we presented has all the necessary resources to reproduce the experiments described. We included a link to the Github repository (https://github.com/ConeasDataset/CONEAS) containing the annotated dataset along with a copy of the ASOHMO dataset with the argumentative information and a python script that can run the experiment described under any setting. We noted, however, that the repository doesn’t contain a README file explaining the script parameters and how to run it so we added it. Any server with the proper hardware should be able to run it. We also created a new repository with the human evaluation of the counter-narratives under https://github.com/ConeasDataset/CounterNarrativesScores. Again, we thank the reviewers for the improvement that the reviewing process has produced in our work.

---

### Official Review · Reviewer_8o9t · 2023-08-02

**Soundness:** 2

**Excitement:**

3: Ambivalent: It has merits (e.g., it reports state-of-the-art results, the idea is nice), but there are key weaknesses (e.g., it describes incremental work), and it can significantly benefit from another round of revision. However, I won't object to accepting it if my co-reviewers champion it.

**Paper Topic And Main Contributions:**

The paper describes an approach to enhance counter-narratives with argumentative information in the social media post to generate more informative responses.

**Questions For The Authors:**

Section 3: who were the annotators? one instance of counter-narrative type per item or more? Not enough detail provided here.

l392: score: why subtract 2 and not map onto a scale from 0 to 1? And how exactly is the score calculated? what are the 'values'? A Likert scale?

Section 4: what were the prompts exactly?

Section 4.3: very small subset of the data for the ground through - how was the data chosen? And who were the analysts? Authors of the paper?

**Reasons To Accept:**

The approach is laudable in itself and it is interesting that in the case of the smaller dataset of Spanish, the rise in performance seems significant.

**Reasons To Reject:**

The paper is very superficial in some places and I'd need more information in the places specified below. I'd also like to see a paragraph of the ethical implications of this work, i.e. a wrong counter narrative could in principle even enhance the impact of the original message.

**Reproducibility:**

3: Could reproduce the results with some difficulty. The settings of parameters are underspecified or subjectively determined; the training/evaluation data are not widely available.

**Reviewer Confidence:**

3: Pretty sure, but there's a chance I missed something. Although I have a good feel for this area in general, I did not carefully check the paper's details, e.g., the math, experimental design, or novelty.

**Typos Grammar Style And Presentation Improvements:**

Typos/grammar/formatting errors are frequent in the paper, examples: newlines in abstract, capitalisation in counter-narrative types, spacing in l287ff, consistent upper-casing in lists, different indentation in Section 4

---

> ### Author Rebuttal · Authors · 2023-08-29
>
> [this is a general comment for all three reviewers] We appreciate the effort made by the reviewers to provide insightful feedback on our work. We understand that it does not exactly conform to the standard experimental paper with standard datasets and standard metrics. We made the decision to address questions that are not straightforward incremental on previous work, and we understand this requires an extra effort from reviewers to identify strong and weak points. We are very thankful for this effort and the very valuable suggestions that they provide. We will be sure to incorporate as much as we can of their work in the final version of this paper.
>
> [This is a shared response for two reviewers] Thank you for pointing out the missing information about the annotation process and the annotators. We will include it in the camera-ready version of the paper, in case it is accepted. The details are as follows:
> Manual generation of CONEAS counter-narratives was conducted by two annotators in three phases. The annotators are both from academic backgrounds, one being a Computer Science PhD student and the other, an undergraduate Philosophy student within the area of Theory of Argumentation. Annotation guidelines were created by the research team before the annotation process began, describing how to use the argumentative analysis of hate tweets to write counter-narratives using different strategies (A, B and C). Both annotators participated on a two hour training to get familiar with the guidelines. Then, during the first phase, each annotator wrote one counter-narrative of each type for the same subsample of 20 tweets in English, then compared the results of their annotation and discussed among each other on how to improve the counter-narrative writing process according to three criteria: a - Adequacy of the counter-narrative as an acceptable, tailored reply for a given hate message; b - Adequacy of the counter-narrative to the specific type of counter-narrative; c - Robustness of the written narrative. This process is challenging since the task is inherently subjective and variability between annotators does not necessarily imply that one of them is wrong, but that guidelines need to cover such variability, without being too lax or too vague. After re-writing and consolidating the guidelines, some counter-narratives were re-written according to the updated guidelines. The second phase of the annotation process consisted in annotating 50 new tweets, and discussing issues that may have arisen and some difficult cases, just to achieve a better alignment between annotators, but without a need to update the guidelines. The third phase consisted on writing counter-narratives for all the rest of the dataset. After this third step, a third annotator reviewed 50 random pairs of hate tweet/counter-narrative using the guidelines as reference, without finding major inadequacies. The first annotator labeled 20% of the total dataset and was not paid (he is part of our research team) while the second annotator labeled the other 80% and was paid 25U$D/hour for 80 hours of work. This project was partially funded by NAACL (we plan to add an acknowledgment and sincere thanks for this in the camera ready version of the paper, if accepted). We want to thank the reviewer for pointing this out. We had originally omitted this information because of space constraints in the paper, but will include it even if it is necessary to add an Appendix.
>
> Concerning the manual evaluation of the automatically generated counter-narratives, this is in a first stage of consolidation. We are aware that evaluation of counter-narratives is a highly complex task, and we intend to carry out a more extensive evaluation with the lessons learned from this first experience. That is why this first phase was small. A small set of guidelines was developed. The annotators were the two annotators who created the manual counter-narratives and a third annotator who is also invested in the project, went through the training with the evaluation guidelines and also participated in the discussion of some test examples to build common ground between annotators. For this work, annotators were paid  the same rate per hour as for the creation of the counter-narratives, and a total of 5 hours was devoted to this evaluation, summing up all 3 evaluators.
>
> Despite the NAACL grant, we have a low budget to carry out this project and had to decide what to fund. We decided to spend most of our budget in obtaining a high quality annotated dataset, instead of investing in obtaining more powerful hardware, because we believe that the quality of annotations makes a bigger impact in this kind of problem. Our main research question was to assess whether this kind of high quality argumentative information did have an impact in the performance of models, more than extensive comparison of models, so we invested most of our little budget in obtaining a high quality dataset. We believe that, to obtain a high quality dataset, one has to employ properly educated annotators, provide them with adequate training and revision processes, and provide high quality working conditions, with good pay and as little as possible time pressure.
>
> Moreover, we also believe that, now that the dataset is publicly available, experiments with larger language models, requiring more powerful computing, can also be carried out by other teams with more computing power available.
>
> [This is a shared response for two reviewers] To our understanding, the paper we presented has all the necessary resources to reproduce the experiments described. We included a link to the Github repository (https://github.com/ConeasDataset/CONEAS) containing the annotated dataset along with a copy of the ASOHMO dataset with the argumentative information and a python script that can run the experiment described under any setting. We noted, however, that the repository doesn’t contain a README file explaining the script parameters and how to run it so we added it. Any server with the proper hardware should be able to run it. We also created a new repository with the human evaluation of the counter-narratives under https://github.com/ConeasDataset/CounterNarrativesScores. Again, we thank the reviewers for the improvement that the reviewing process has produced in our work.
>
> [This is a shared response for the three reviewers] Thanks to your comments we understand that the values of the score are difficult to interpret. In the final version of the paper we will provide the algorithm showing how the score is calculated, and will illustrate it with a couple of examples in an Appendix. We believe that a formula will not make the meaning of the score more clear, because it is a sum of the score obtained for each of the four dimensions making up the final score, which are each a Likert scale ranging from 1 to 3 in three cases and 0 to 3 in the fourth, with a maximum possible score of 12. Moreover, one of those dimensions, “Informativeness” only scores more than 0 if the dimension of “Stance” scores at the maximum of 3.
>
> Normalizing the final score to a range from 0 to 1 is definitely a suggestion that we will take, as it makes the score more interpretable. However, seeing the difficulties in interpreting the metric, we will also provide the detail of the score obtained by each approach for each of the four dimensions evaluated, as we believe this provides more transparency for the way the final score is built up.
>
> Concerning the illustration of prompts, we included an example of the prompt used on the appendix. We don’t know if the reviewer couldn’t find the Appendix or if the provided prompt is not clear enough. We didn’t include it in the main sections of the paper due to space constraints but we will make an effort to include it in the extra page for the main body of text for the camera ready version. We will also include more examples in the Appendix.
>
> The training, tuning and test partitions were determined by the original ASOHMO test, taking the proposed partitions as standard. For manual evaluation of the generated counter-narratives we selected a random sample of 20 tweets in English and 10 tweets in Spanish for most experimental settings, involving all types of counter-narratives. We couldn’t do a more extensive evaluation because of budget and time constraints, but are currently developing a new set of guidelines with lessons learned from this first phase of the project and will be carrying out a more extensive evaluation with these improved guidelines.
>
> With respect to the possibility that a wrong counter narrative could be harmful, contributing to the hate in the original message, we totally agree with the reviewer that it must be signaled as a high-cost error. We are not very experienced in writing Ethical Considerations sections and some aspects have skipped our attention. In the final version of the paper, we will include this within the ethical considerations as an error to be handled with special care. Our recommendation is that counter-narrative generation should not be used as a stand-alone tool, but rather as an aid for human experts to moderate the conversation in social media and other sites of such conflicts.

---

### Official Review · Reviewer_4FEa · 2023-08-04

**Soundness:** 4

**Excitement:**

4: Strong: This paper deepens the understanding of some phenomenon or lowers the barriers to an existing research direction.

**Missing References:**

Belz, Howcroft etc regarding evaluative criteria for human evaluation

**Paper Topic And Main Contributions:**

The authors deal with highly-targeted fine-tuning for hate speech counter-narrative generation in social media. Counter-narrative generation is the opposite to mere detection and blockage of hate speech in social media, which leads to overblocking or censoring.
The authors hypothesize that additional information about argumentative structure of the hate speech instances may improve the counter-narration.

The authors manually constructed counter-narratives for an existing corpus, dubbing their bilingual dataset CONEAS,  containing 4 different narratives per argumentative tweet. They trained T5 derived models on the data.

Several experiments were conducted.

For controlling the influence of the size of the models, the authors tested several sizes of FlanT5.
After preliminary testing, larger models did not perform significantly better such that tests were aborted for energy-saving reasons.

Furthermore they examined the qualitative difference between fine-tuning and fewshot learning

Given their main hypothesis about argumentative information enrichment, they perfomed an ablation study on the argumentation information present in their data (ASOHMO)

They also investigated whether the type of the counter-narrative (4 types) has influence.

For evaluation, they manually assessed the criteria offensiveness, Informativeness and felicity.

Interestingly, the authors find that on English data, argumentative information enrichment causes the larger model to perform worse than the smaller model!!
This is an instance of a pointer that linguistic engineering reduces model size need.
The main finding of evaluation is that finetuning a smaller model on a small set of high quality data improves generation of counter-narratives above the level of larger models. Furthermore some kind of argumentative imformation may improve informativity of generated counter-narratives.

**Reasons To Accept:**

The authors invested much effort into building the necessary data, their hypotheses are perfectly adequate and the presented methodology is sound. Some findings even bear evidence beyond their topic of generating counter-narratives, namely that enriching input data with linguistic / argumentative information may improve small models above the level of larger models. Their critical position against BLEU, BERTScore, Rogue etc. is fully justified and its been a pleasure to note that the authors did not report any uninformative scores - this is quite rare in times where plain, sometimes unexplainable differences in BLEU scores have become the stereotypical evaluation method.

**Reasons To Reject:**

One weakness, two sides of the same medal. The criteria you use for human evaluation (Informativeness, felicity etc.) are not exclusive to this subfield of generation, but are also widely discussed in the generation literature. It would have been a good idea to discuss or at least mention some of the unification ideas of Belz et al. or Howcroft et al. that is supposed to introduce more objectiveness into the terms / concepts, which otherwise remain opaque. (BUT actually, this is not the authors' fault - it is an effort that the whole NLG community has to make in order to define fundamentally improved evaluation metrics.)

The only other issue is that the conclusions one can draw from the experiments are not yet as clear as one could wish, but hopefully further research will result in a clearer picture (again this is not a conceptual error in the models or experiments - just an inherent issue with empirical studies.).

**Reproducibility:**

4: Could mostly reproduce the results, but there may be some variation because of sample variance or minor variations in their interpretation of the protocol or method.

**Reviewer Confidence:**

4: Quite sure. I tried to check the important points carefully. It's unlikely, though conceivable, that I missed something that should affect my ratings.

---

> ### Author Rebuttal · Authors · 2023-08-29
>
> [this is a general comment for all three reviewers] We appreciate the effort made by the reviewers to provide insightful feedback on our work. We understand that it does not exactly conform to the standard experimental paper with standard datasets and standard metrics. We made the decision to address questions that are not straightforward incremental on previous work, and we understand this requires an extra effort from reviewers to identify strong and weak points. We are very thankful for this effort and the very valuable suggestions that they provide. We will be sure to incorporate as much as we can of their work in the final version of this paper.
>
> Concerning the fact that one cannot draw very strong conclusions from the experiments presented here, we agree with the reviewer that the experiments are small and the agreement figures leave room for improvement. As we explain below, we have been working to improve that. However, we believe that these results are worth sharing with the community as they can encourage more research in injecting high-quality information into large language models. Also, they are an illustrating case of a successful combination of symbolic and subsymbolic approaches, moreover, in a task that has been very challenging for long. We will make an effort to put more emphasis on these conclusions in the final version of the paper.
>
> We are very grateful for the insights from the NLG community with respect to evaluation! Indeed, we were only distantly aware of work in that area, and for the evaluation provided in this work we mainly focused on proposals from argumentation evaluation. We find that evaluation of open-ended tasks is mainly carried out with deeply inadequate, n-gram based methods, which obscures more insightful approaches. We found in ACL 2023 a paper on counter-speech generation [1] that is based solely on such n-gram- or embedding-based automatic metrics of evaluation, as those criticized in [2] (thanks for the pointer!).
>
> Since we got the reviews, we have been skimming through the work of Anja Belz and collaborators and Howcroft and collaborators, and indeed they have provided concepts and operationalizations that are very useful for a second version of our evaluation protocol. Indeed, the evaluation protocol presented in this paper was our first try for the manual evaluation of counter-narratives. We think this protocol enabled a first glimpse on the quality of the obtained counter-narratives, and it is very valuable for us, and hopefully for the community, to show that high-quality argumentative information has a positive impact in improving automatically generated counter-narratives. As such, we believe this is a valuable result to be shared with the community. However, the agreement figures were worrying.
>
> Since we obtained the agreement figures, we have been working on how to improve them, and have come to the same conclusion as the reviewer: that we need to zoom into the dimension we dubbed “Informativeness” to make room for the acceptability of arguments and better capture the persuasiveness of the argument, as developed in the recent analysis by Wagemans and Hinton [3]. We have just finished creating a second, improved version of the guidelines for manual evaluation of counternarratives where Informativeness will be replaced by four dimensions, each representing one of the Gricean maxims: Quantity, Quality, Manner and Relevance. We expect that this granularity will allow for a more accurate evaluation rubric, which will hopefully result in higher agreement between annotators. We plan to carry out an exploratory evaluation of the size of the evaluation presented in this paper and, after that, a more extensive one.
>
> Concerning the pointers provided by the reviewer, we will definitely incorporate the concepts provided by the NLG evaluation people, to try to make the concepts in our current and future evaluation schemes more transparent for the community. Additionally, we found the Human Evaluation Datasheet [4] very useful to systematize the analysis of our task in order to obtain a better evaluation protocol, and will describe some of how it helped in the final version of the paper. Also the analysis and proposal in [5] has been inspiring and will incorporate some of it, together with the analysis in [6]. We will also be incorporating some of the general insights of [7], which we find very cogent and valuable to share in the context of this work.
>
> Again, we want to thank the reviewer for these pointers that will definitely help to make a more insightful, wider evaluation in the future.
>
> References:
>
> [1] Rishabh Gupta, Shaily Desai, Manvi Goel, Anil Bandhakavi, Tanmoy Chakraborty, Md. Shad Akhtar. 2023. Counterspeeches up my sleeve! Intent Distribution Learning and Persistent Fusion for Intent-Conditioned Counterspeech Generation
> https://aclanthology.org/2023.acl-long.318/
>
> [2] Ehud Reiter and Anja Belz. 2009. An Investigation into the Validity of Some Metrics for Automatically Evaluating Natural Language Generation Systems
> https://aclanthology.org/J09-4008.pdf
>
> [3] Jean H.M. Wagemans and Martin Hinton. 2022. How persuasive is AI-generated argumentation? An analysis of the quality of an argumentative text produced by the GPT-3 AI text generator
> Argument & Computation
>
> [4] Anastasia Shimorina and Anya Belz. 2022. The Human Evaluation Datasheet: A Template for Recording Details of Human Evaluation Experiments in NLP
> https://aclanthology.org/2022.humeval-1.6.pdf
>
> [5] Rudali Huidrom and Anya Belz. 2002. A Survey of Error Annotation Schemes for Human and Machine Generated Text
> https://aclanthology.org/2022.gem-1.33.pdf
>
> [6] Anya Belz, Simon Mille, David M. Howcroft. 2020. Disentangling the Properties of Human Evaluation Methods: A Classification System to Support Comparability, Meta-Evaluation and Reproducibility Testing
> https://www.aclweb.org/anthology/2020.inlg-1.24/
>
> [7] Non-Repeatable Experiments and Non-Reproducible Results: The Reproducibility Crisis in Human Evaluation in NLP
> Anya Belz, Craig Thomson, Ehud Reiter, Simon Mille

---

### Meta-Review · Area_Chair_Ln4m · 2023-09-18

**Recommendation:** 3

**Metareview:**

This work tackles counter-narrative generation for hate speech on social media. Unlike existing works largely leveraging the hate speech itself, this work uses argumentative structures from the context. Results show that adding argumentative structures helps boost the results even in a few-shot setting.

Reviewers appreciated the importance of the task, effective use of argument information and the two datasets constructed for this study.

At the same time, reviewers are concerned about the limited details on the annotation process, the discussion on evaluation, and the limited experiments conducted to underpin the main claims of the paper. More discussions on the annotation and evaluation will be helpful to the community.

---

### Decision · Program_Chairs · 2023-10-07

**Decision:**

Accept-Findings

**Comment:**

This work tackles counter-narrative generation for hate speech on social media. Unlike existing works largely leveraging the hate speech itself, this work uses argumentative structures from the context. Results show that adding argumentative structures helps boost the results even in a few-shot setting.

Reviewers appreciated the importance of the task, effective use of argument information and the two datasets constructed for this study.

At the same time, reviewers are concerned about the limited details on the annotation process, the discussion on evaluation, and the limited experiments conducted to underpin the main claims of the paper. More discussions on the annotation and evaluation will be helpful to the community.